# Effect of neutering timing in relation to puberty on health in the female dog–a scoping review

**Rachel Moxon**[1]*, **Gary C. W. England**[1], **Richard Payne**[1], **Sandra A. Corr**[2], **Sarah L. Freeman**[1]

**1** School of Veterinary Medicine and Science, University of Nottingham, Sutton Bonington, Leicestershire, United Kingdom, **2** School of Biodiversity, One Health and Veterinary Medicine, College of Medical, Veterinary and Life Sciences, University of Glasgow, Glasgow, United Kingdom

* rachel.moxon@nottingham.ac.uk

**Data Availability Statement:** All relevant data are within the manuscript and its Supporting Information files.

## Abstract

### Background

Effects of neutering on bitch health have been reported, and are suggested to relate to bitch age at the time of neutering for some diseases. However, variation between published studies in terms of study populations and methodologies makes comparison and consolidation of the evidence difficult.

### Objective

A scoping review was designed to systematically search the available literature to identify and chart the evidence on the effect of neutering timing in relation to puberty on five health outcomes: atopy, developmental orthopaedic disease (DOD), neoplasia, obesity and urogenital disease.

### Design

A protocol was registered, and literature searches were conducted in CAB Abstracts, Medline and Web of Science. Studies were reviewed against inclusion criteria. Data on study and population characteristics and health outcomes were charted for the final included studies.

### Results

A total of 1,145 publications were reviewed across all five searches; 33 were retained for inclusion and charting. Only six of the 33 studies categorised the timing of surgical neutering as prepubertal or post-pubertal; one investigating mammary neoplasia and the other five, urogenital disease, commonly urinary incontinence. No studies were identified that examined the impacts of neutering bitches before or after puberty on atopy, DOD or obesity. One study considered bitches that were pre or post-pubertal at the time of the first treatment with deslorelin acetate for oestrus suppression and 26 examined the effects on health related to age, rather than pubertal status, at neutering.

**Funding:** The author(s) received no specific funding for this work.

**Competing interests:** The authors have declared that no competing interests exist.

## Conclusion

This scoping review suggests that robust evidence to support veterinarians, those working with dogs and dog owners when discussing the timing of neutering relative to puberty does not yet exist. The impact of neutering *before or after puberty* on atopy, DOD, neoplasia, obesity and urogenital disease in female domesticated dogs remains unclear.

## Introduction

The impact of neutering on female dog health has been studied and summarised in many narrative and systematic reviews [1–9]. Both positive and negative effects on bitch health have been reported, and these are suggested to be associated with age at neutering for some diseases [10–18]. However, comparing and consolidating evidence from different studies is challenging due to variations in study populations and methodological approaches, along with the multifactorial nature of the diseases studied. The direction of the effect can also be inconsistent depending on the disease studied. For example, earlier neutering has been associated with an increased risk of cranial cruciate ligament (CCL) rupture (neutering at less than 12 months of age [13–15, 19]), but a decreased risk of certain cancers such as haemangiosarcoma (HSA), mammary neoplasia and mast cell tumours (MCT) (neutering before 2.5 years of age [20, 21]; neutering at less than 12 months of age [13, 14]). For some diseases, such as urinary incontinence (UI), the association with age at neuter is unclear: Lutz *et al.* [22] reported an increased risk for bitches neutered at earlier ages, while Reichler *et al.* [23] reported the opposite, and others failed to find a significant association [24–26]. Breed differences are also reported for certain diseases, which further complicates interpretation of the findings [16, 17].

Two systematic reviews conducted in 2012 [1, 2] investigated the effect of neutering and age at neutering (including studies that considered pubertal status at neutering) on mammary tumours and UI. Only weak associations were identified, and the authors highlighted risks of bias in the published studies. In other narrative reviews considering neutering timing in relation to puberty, mammary tumours are the most frequently mentioned disease, with a suggested protective effect of earlier neutering [27–33]. Most reviews reference the same work by Schneider *et al.* [20] that identified a reduced risk of histologically malignant mammary neoplasms (adenocarcinomas and mixed mammary tumours) for bitches neutered before the first oestrus. However, this study was suggested to be at moderate risk of bias by Beauvais *et al.* [1]. Other narrative reviews have suggested variable impacts on UI based on individual study findings [30] while some suggest that the evidence relating to the impacts on UI is inconclusive [34]. Neutering before puberty has been suggested to impact growth and vulva development [29, 34–36], leading to epiphyseal fractures and urogenital disease, however the work of Salmeri *et al.* [10], which did not consider neutering before or after puberty, is commonly referenced to support these suggestions. Inferring pubertal status based on studies that have used age at neutering is potentially flawed due to the variation in age at onset of puberty between dogs (six to 18 months; [37]), the fact that actual puberty (first oestrus) is not recorded, and that the age groups in studies are not designed to adequately distinguish between these groups.

To the authors' knowledge, no systematic reviews or research syntheses have yet been published on the impact of neutering bitches before or after puberty on atopy, DOD, neoplasia (other than mammary tumours), or obesity. Additionally, since the two systematic reviews on the impact of neutering on mammary tumours and UI [1, 2], further studies have reported results which would benefit from review. Due to the potential for the timing of neutering in relation to puberty to impact various aspects of long-term health in female dogs, a scoping

review was designed. Scoping reviews can provide a broad overview of literature relating to a topic, are a useful first step in collating the evidence and can be used prior to a systematic review [38–41].

The aim of this study was to conduct a scoping review to identify and chart the current evidence on the effect of the timing of neutering in relation to puberty on the health of female domesticated dogs, and to highlight areas where knowledge is lacking.

## Materials and methods

### Protocol and registration

A search was conducted for existing scoping and systematic reviews on the impact of neutering bitches before and after puberty on atopy, DOD, neoplasia, obesity and urogenital disease on JBI Evidence Synthesis, PROSPERO, PubMed and VetSRev on 07 September 2023. Two systematic reviews were identified, however these primarily examined the evidence related to neutering age and not pubertal status at neutering [1, 2].

A Preferred Reporting Items for Systemic reviews and Meta-Analyses Extension for Scoping Reviews (PRISMA-ScR; [42]) was used for this scoping review. The scoping review protocol was developed before data extraction and was registered with the Open Science Framework [43]. The project was reviewed and approved by the Ethics Committee, School of Veterinary Medicine and Science, University of Nottingham.

### Search strategy

Five primary literature searches were conducted on 17/05/2023 in three scientific databases; CAB Abstracts (1910 to present), Medline (1910 to present) and Web of Science (1950 to present) [44]. Search terms were developed and reviewed by a team of six people, including the authors and a librarian (see S1 File). Additionally, BestBETs, RCVS Knowledge Summaries and VetSRev were searched. Papers were also included that were found in a previous literature search by the primary author but were not identified in the primary literature searches.

### Study selection

The search results were exported into EndnoteX8 (Thomson Reuters). Duplicate records were identified and deleted before publications were assessed against the inclusion criteria (see S2 File). A three-stage systematic review and exclusion process was completed independently by two researchers involving 1) review of publication titles, 2) review of publication abstracts and 3) review of the full-text for the remaining publications. At title and abstract review, ambiguous publications were retained for the next review stage. Publications where abstracts could not be identified were retained for full-text review. For publications where full-text was not available, individual authors and journals were contacted to attempt to obtain the full-text. A study was excluded if English language full-text was not available from University of Nottingham libraries or e-libraries, from free online Open Access and legal deposit libraries, from online searches, or from direct contact with authors or journals.

### Charting process

The remaining final full-text publications were assessed independently by two researchers to ensure eligibility for inclusion. Each search was described separately and then characteristics and relevant information were extracted and presented.

**Data extraction.** Following full text review, key information was extracted from each of the 33 retained publications using a standardised form (see S3 File) and tabulated. Study

characteristics extracted and charted were: author, publication year, data source or country, study design, sample size, study aims, details of the intervention, health outcomes, how outcomes were measured and whether dogs were randomly or retrospectively allocated to study groups. Study population characteristics extracted and charted were: how the sample was sourced, dog breed, age and sex, whether dogs neutered for a related health problem were excluded, whether previous breeding history was considered and how puberty was defined. This was carried out independently by one researcher.

**Summarising the available evidence.** The 33 publications were assessed by two researchers. Significant effects of timing of neutering on each health outcome were reported for either bitches or dogs and bitches (where studies did not present results for sexes of dogs separately). Studies that found no significant effects, reported significant effects only in male dogs, or made no comparisons between different neuter age groups were not included in the charting tables. A summary column was included to show the reported impact of neutering at certain times more clearly, including direction of any effect, for each health outcome included from each study.

## Results

### Study selection

**Atopy.** The primary literature search identified 50 publications following the removal of duplicates. Thirty-five were excluded on title review, seven were excluded following abstract review and seven were excluded on full-text review, including six for concept and one that was not available in English. Therefore, one publication remained for assessment (Fig 1A).

**Developmental orthopaedic disease.** The primary literature search identified 74 publications following the removal of duplicates. Twenty-eight were excluded on title review, 29 were excluded following abstract review and four were excluded on full-text review, including one that was not available in English, one that did not have full text available, one for concept and one for study design. Therefore, 13 publications remained for assessment (Fig 1B).

**Neoplasia.** The primary literature search identified 513 publications following the removal of duplicates. Three hundred and forty were excluded on title review, 137 were excluded following abstract review and 25 were excluded on full-text review, including seven that were not available in English, three that did not have full text available, five for publication type and 10 for concept. Therefore, 11 publications remained for assessment (Fig 1c).

**Obesity.** The primary literature search identified 182 publications following the removal of duplicates. One hundred and seven were excluded on title review, 57 were excluded following abstract review and 12 were excluded on full-text review including one that was not available in English, two for publication type and nine for concept. Therefore, six publications remained for assessment (Fig 1D).

**Urogenital.** The primary literature search identified 326 publications following the removal of duplicates. Two hundred and twenty-five were excluded on title review, 63 were excluded following abstract review and 20 were excluded on full-text review, including two that were not available in English, two that did not have full text available, six for publication type and 10 for concept. Therefore, 18 publications remained for assessment (Fig 1E).

### Study characteristics

There were 33 individual papers identified in total across the five independent searches that were included in this review, some were identified in more than one of the searches (Table 1). Papers were published between 1969 and 2023 and were most commonly from the USA/North America (21/33 studies), including all of the papers identified for atopy, DOD and obesity, and

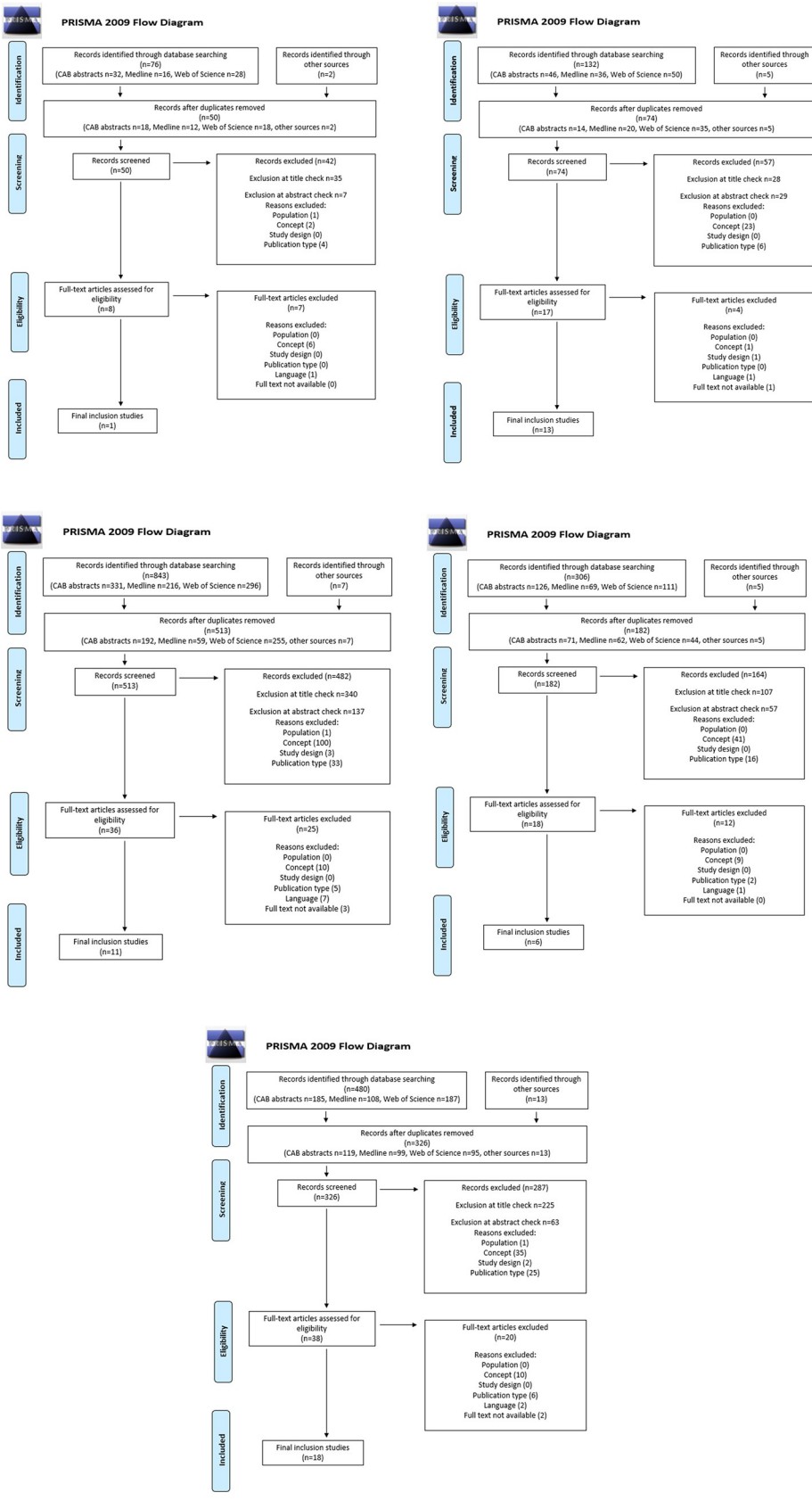

**Fig 1. a.** The number of publications that were identified, reviewed, and excluded from the scoping review on the effect of neutering timing on atopy in female dogs. **b.** The number of publications that were identified, reviewed, and excluded from the scoping review on the effect of neutering timing on developmental orthopaedic disease in female dogs. **c.** The number of publications that were identified, reviewed, and excluded from the scoping review on the effect of neutering timing on neoplasia in female dogs. **d.** The number of publications that were identified, reviewed, and excluded from the scoping review on the effect of neutering timing on obesity in female dogs. **e.** The number of publications that were identified, reviewed, and excluded from the scoping review on the effect of neutering timing on urogenital disease in female dogs.

9/11 for neoplasia. Dogs were most commonly grouped retrospectively into neuter groups based on age (23/33 studies) or pubertal status (5/33 studies) at neutering. Two studies randomly allocated dogs to neutering age groups and two studies grouped dogs based on age when neutered. Twenty-two studies did not report the type of neutering surgery performed. One study included bitches that had received deslorelin acetate implants for oestrus suppression. There were 27 different health outcomes examined and these were most frequently assessed by examining veterinary records (15/33) or by owner (8/33) or veterinarian questionnaires (4/33). Four studies used direct observations. Three owner questionnaire studies also used additional collection methods: data from veterinary records, assistance from the veterinarian or veterinarian questionnaires (Table 1).

## Study population characteristics

Eighteen of the 33 studies included dogs of various breeds and usually did not report individual breeds. Fifteen studies included only bitches, 12 of these investigated urogenital outcomes, two neoplasia, and one urogenital and neoplasia. Eighteen studies included dogs and bitches, nine of these did not report results for each sex by neuter group independently, one reported some results by sex and eight did report results for male and female dogs separately. All papers identified for atopy, DOD and obesity were based on age not pubertal status at neutering. One paper for neoplasia and six for urogenital disease examined data by pubertal status at neutering. One further study aimed to examine data by pubertal status at the time of neutering however a lack of data prevented this analysis [54]. One study that reported results for bitches neutered before or after puberty had potential problems associated with grouping of bitches, with some older bitches reported to be classed as prepubertal (up to 1.4 years of age) and some younger bitches included in the post-pubertally neutered group (minimum 0.3 years of age) [22]. Only six studies examined data for bitches that were surgically neutered before or after puberty. One of these compared results for bitches neutered before puberty in the study to those for bitches neutered after puberty from another study [56]. One study examined data for bitches treated with deslorelin acetate to suppress oestrus, however data were not analysed statistically, and numbers were too small for comparative analysis. The remaining 26 studies either did not mention puberty (22/26), did not have the data available to examine puberty (1/26) or based conclusions regarding puberty on age at neutering (2/26) (Table 2).

## Charting the studies against the aims of the scoping review

Thirteen different health outcomes were reported to be significantly affected by timing of neutering under the five health categories investigated (Table 3). The only study identified that examined atopy did not report results in relation to to pubertal status or age at neuter [57]. Two studies reported no impact on DOD (CCL rupture [59]; overall incidence and HD [11]). One study reported no impact on neoplasia (mammary neoplasia [12]). Three studies reported no impact on obesity (weight gain and back fat measurements [10]; owner perception of their dog's body weight [11]; overweight and obese based on body condition score [50]). Five

**Table 1. The study characteristics for the publications identified for inclusion in the scoping review of the literature undertaken on the effect of timing of neutering on atopy, developmental orthopaedic disease (DOD), neoplasia, obesity and urogenital disease in female domesticated dogs.**

| Author and year | Source / country | Study design | Sample size | Aims* | Intervention | Relevant health outcomes* | Outcome measures | Allocated randomly to neutering groups or retrospectively grouped | Comparisons reported between dogs neutered in different groups or just to the entire group? |
|---|---|---|---|---|---|---|---|---|---|
| Beaudu-Lange et al., 2021 [45] | France | RC | 599 females | Determine the frequency of reproductive disorders over the lifetime of female dogs according to neuter status | Ovariectomy–early neuter (<2 years of age), late neuter (>2 years of age), entire | 1) Neoplasia (mammary tumour—including benign) 2) Urogenital disease (UI) | Veterinary records | Retrospectively grouped | Comparisons between early and late-neuter groups reported for mammary tumours only, not for UI |
| Brandli et al., 2021 [46] | Switzerland | RC | 32 females | Evaluate the side effects of repeated deslorelin acetate application for sustained gonadal suppression | 4 bitches first implant before puberty, 28 bitches after puberty, deslorelin acetate implant for oestrus suppression | Urogenital disease (UI) | Owner questionnaires | Retrospectively grouped | Descriptive comparisons reported between dogs first treated before or after puberty, no statistical analysis |
| Byron et al., 2017 [47] | USA | CC | 163 UI, 193 control | Examine the impact of age at OVH and weight at presentation for incontinence on the hazard of USMI | OVH | Urogenital disease (USMI stated, however UI and incontinence also used interchangeably throughout so unclear) | Data supplied by veterinarians | Retrospectively grouped | Relationships with age at neuter reported |
| Cooley et al., 2002 [48] | USA and Canada | RC | 683 dogs (389 female) | Test the hypothesis that endogenous sex hormones significantly influence bone sarcomagenesis | Neutering, surgical method not stated | Neoplasia (appendicular bone sarcoma) | Owner questionnaires completed with veterinary assistance | Retrospectively grouped | Between age group comparisons reported |
| de Bleser et al., 2011 [25] | UK | CC | 202 cases, 168 controls | Estimate the strength of association between early spaying and acquired USMI | Neutering, surgical method not stated. 59 bitches neutered <6 months of age and 255 >6 months of age. 100 bitches neutered before and 229 after puberty | Urogenital disease (acquired USMI) | Owner questionnaires | Retrospectively grouped | Between age group and pre/post-pubertal comparisons reported |

(*Continued*)

**Table 1.** (Continued)

| Author and year | Source / country | Study design | Sample size | Aims* | Intervention | Relevant health outcomes* | Outcome measures | Allocated randomly to neutering groups or retrospectively grouped | Comparisons reported between dogs neutered in different groups or just to the entire group? |
|---|---|---|---|---|---|---|---|---|---|
| Duerr *et al.*, 2007 [49] | North America | RC | For age at neuter analysis: 80 dogs, number of females not reported | Identify risk factors for excessive tibial plateau angle (TPA) in large-breed dogs with CCLD | Neutering grouped as <6 (n = 31) or >6 months (n = 49), number of male and female in each neuter age group and surgical method not stated | DOD–TPA in dogs with CCL disease | Measurements of TPA from radiographs for cases and from dogs' medical records for controls | Retrospectively grouped | Between age group comparisons reported |
| Ekenstedt *et al.*, 2017 [19] | North America | RC | 174 cases, 139 controls (108 neutered female dogs) | Examine the association between EIC and CCLR and examine aspects, such as sex and neutering | Neutering grouped as ≤1 (83 female) or >1 year (25 female), surgical method not stated | DOD–CCL rupture | Most cases defined by surgical confirmation, others defined via physical exam, stifle palpation, and radiographs. Controls—no history of pelvic limb lameness or stifle surgery, no abnormalities in either knee via orthopaedic examination | Retrospectively grouped | Between age group comparisons reported |
| Forsee *et al.*, 2013 [26] | UK | RC | 566 female | Identify the prevalence and determine associations of AUI with specific variables in OVH bitches | OVH. 109 bitches neutered <6 months of age, 355 neutered 6 to <18 months, 102 neutered ≥18 months | Urogenital disease (AUI) | Owner questionnaires | Retrospectively grouped | Between age group comparisons reported |
| Hart *et al.*, 2014 [14] | USA | RC | Golden Retrievers = 1,015 (472 females, 306 neutered). Labradors = 1,500 (692 females, 347 neutered) | Explore the effects of neutering on joint disorders and cancers in Labradors and Golden Retrievers | Neutering, surgical method not stated. Age grouped as <6 months, 6 to 11 months, 12 to <24 months, and 2 to 8 years. Number of dogs neutered at each age not clear | DOD (HD, ED, CCL) Neoplasia (LSA, HSA, MCT, MC) | Veterinary records | Retrospectively grouped | No comparisons reported between dogs neutered at different ages |

*(Continued)*

**Table 1.** (Continued)

| Author and year | Source / country | Study design | Sample size | Aims* | Intervention | Relevant health outcomes* | Outcome measures | Allocated randomly to neutering groups or retrospectively grouped | Comparisons reported between dogs neutered in different groups or just to the entire group? |
|---|---|---|---|---|---|---|---|---|---|
| Hart et al., 2016 [15] | USA | RC | 705 male (245 neutered), 465 females (293 neutered) | Examine relationships between age at neutering and joint disorders and cancers | Neutering, surgical method not stated. Age grouped as <6 months, 6 to 11 months, 12 to <24 months, and 2 to 8 years. Number of dogs neutered at each age not clear | DOD (HD, ED, CCL) Neoplasia (LSA, HSA, MCT, MC) Urogenital disease (UI) | Veterinary records | Retrospectively grouped | No comparisons reported between dogs neutered at different ages |
| Hart et al., 2020a [16] | USA | RC | Over 15,000 dogs, neutered and entire, male and female, exact numbers unclear | Provide information on breed-specific differences and guidelines for neutering ages considering long-term health risks of neutering. Document breed-specific differences in the increases in some cancers associated with neutering | Neutering, surgical method not stated. Age grouped as <6 months, 6 to 11 months, 12 to <24 months, and 2 to 8 years. Number of dogs neutered at each age not clear | DOD (HD, ED, CCL) Neoplasia (LSA, HSA, MCT, MC, OSA) Urogenital disease (UI) | Veterinary records | Retrospectively grouped | No comparisons reported between dogs neutered at different ages |
| Hart et al., 2020b [17] | USA | RC | 3,139 (441 entire female, 1159 neutered female) | Provide information for mixed breed dogs on the best age to neuter considering the risks of joint disorders and cancers | Neutering, surgical method not stated. Age grouped as <6 months, 6 to 11 months, 12 to <24 months, and 2 to 8 years. Number of dogs neutered at each age not clear | DOD (HD, ED, CCL) Neoplasia (LSA, HSA, MCT, MC, OSA) Urogenital disease (UI) | Veterinary records | Retrospectively grouped | No comparisons reported between dogs neutered at different ages |

(*Continued*)

**Table 1.** (*Continued*)

| Author and year | Source / country | Study design | Sample size | Aims* | Intervention | Relevant health outcomes* | Outcome measures | Allocated randomly to neutering groups or retrospectively grouped | Comparisons reported between dogs neutered in different groups or just to the entire group? |
|---|---|---|---|---|---|---|---|---|---|
| Howe *et al.*, 2001 [11] | USA | PC | 269 (153 female) | Determine long-term results and complications of neutering performed at an early age (prepubertal) or at the traditional age in dogs | OVH or castration. 94 dogs neutered ≥24 weeks of age and 175 neutered <24 weeks of age. Sex ratio in each neuter age group not reported | DOD (overall incidence of musculoskeletal disease and HD) Obesity (owner perception of dog's bodyweight) Urogenital disease (urinary problems) | Owner telephone questionnaires 41–64 months after adoption. Contact with veterinarians for any unclear diagnoses | Grouped based on age at surgery | Between age group comparisons reported |
| Lefebvre *et al.*, 2013 [50] | USA | RC | 1,930 neutered and 1,669 entire dogs | Determine whether gonadectomy or age at gonadectomy was associated with the risk of becoming overweight | Neutering, surgical method not stated. 782 neutered ≤6 months of age (454 female), 861 at >6 months to ≤1 year of age (469 female), 287 at >1 to ≤5 years of age (138 female) | Obesity (overweight or obese) | Body condition score 4 or 5 out of 5 in veterinary records | Retrospectively grouped | Between age group comparisons reported |
| Lutz *et al.*, 2019 [22] | Switzerland | RC | 131 pair-matched dogs | Investigate the risk factor "time of spaying relative to the onset of puberty" on AUI | Neutering, surgical method not stated | Urogenital (AUI) | Owner questionnaire | Retrospectively grouped | Before and after puberty compared |
| Moxon *et al.*, 2023 [51] | UK | PC | 306 female | Compare vulval size and appearance at 17 months of age for bitches neutered before or after puberty | OVH before (n = 155) or after (n = 151) puberty | Urogenital (vulval size and appearance–recessed and juvenile vulva, vulval discharge, perivulval folds and skin changes, % dorsal fold coverage) | Veterinary assessment and assessment of digital vulval images | Randomly allocated | Before and after puberty compared |
| Palerme *et al.*, 2021 [52] | USA | CS | 250 females, 216 neutered | Assess the prevalence and severity of recessed vulvas and characterise differences between bitches with and without recessed vulvas | OVH. Number in each neuter age group not reported | Urogenital (recessed vulva) | Direct observation, percentage of vulvar skin coverage scored out of 8 recessed vulva = score ≥7 | Retrospectively grouped | Between age group comparisons reported |

(*Continued*)

**Table 1.** (Continued)

| Author and year | Source / country | Study design | Sample size | Aims* | Intervention | Relevant health outcomes* | Outcome measures | Allocated randomly to neutering groups or retrospectively grouped | Comparisons reported between dogs neutered in different groups or just to the entire group? |
|---|---|---|---|---|---|---|---|---|---|
| Pegram *et al.*, 2019a [53] | UK | RC | 23,499 neutered and 49,472 entire females | Explore association between neuter status and age at neuter with early-onset UI | Neutering, surgical method not stated. Neutering age grouped as <6 (n = 3,418), 6 to <12 (n = 10,543), 12 to <24 (n = 5,426) and ≥24 months (n = 4,112) | Urogenital (UI) | Veterinary records | Retrospectively grouped | Between age group comparisons reported |
| Pegram *et al.*, 2019b [54] | UK | CC | 106 case and 463 control bitches with age at neuter data | Explore the association between UI and neuter status, age at neuter and neuter relative to first oestrus | Neutering, surgical method not stated. Neutering age grouped as <6 (n = 68), 6 to <12 (n = 159), 12 to <24 (n = 144) and ≥24 months (n = 198) | Urogenital (UI) | Veterinary records | Retrospectively grouped | Between age group comparisons reported |
| Salmeri *et al.*, 1991 [10] | USA | PC | 32 (17 female) | Determine how weight gain and body fat was affected by neutering at 7 weeks versus 7 months of age | Females neutered via OVH. Seven-week neuter: 7 male 7 female 7-month neuter: 4 male 4 female Entire: 4 male 6 female | Obesity (weight gain and body fat) | Weight gain–monthly bodyweight measurements Body fat–back fat depth surgically at 14 months of age | Randomly allocated | Between age group comparisons reported |
| Schneider *et al.*, 1969 [20] | USA | CC | 87 female case-control matches | To measure the effect on canine mammary cancer risk of various factors including neutering | Neutering, surgical method not stated. Neutered before first oestrus, after one or after two oestrous cycles | Neoplasia (mammary) | Neoplasia based on tumours submitted to neoplasm registry. Demographic data from owner interviews and veterinary records | Retrospectively grouped | Before and after puberty compared |

(*Continued*)

Table 1. (Continued)

| Author and year | Source / country | Study design | Sample size | Aims* | Intervention | Relevant health outcomes* | Outcome measures | Allocated randomly to neutering groups or retrospectively grouped | Comparisons reported between dogs neutered in different groups or just to the entire group? |
|---|---|---|---|---|---|---|---|---|---|
| Simpson, 2017 [55] | USA | PC | 3,044 dogs (number of females not reported) | Examine whether age at gonadectomy is associated with overweight / obesity and chronic non-traumatic orthopaedic injuries | Neutering (gonadectomy) at <6 months, 6 to 12 months, >12 months or entire. Surgical method and number of dogs in each neuter group not reported | DOD (first occurrence of veterinarian-reported CCL rupture or osteoarthritis) Obesity (first occurrence of veterinarian-reported overweight or obese) | Annual data collection from owners and veterinarians Overweight / obesity–Purina Body Condition Score scale of >6/9 | Retrospectively grouped | No comparisons reported between dogs neutered at different ages |
| Simpson et al., 2019 [18] | USA | PC | 2,764 or 2,754 dogs–unclear (number of females not reported) | Examine associations between gonadectomy and overweight / obesity and orthopaedic injuries | Neutering (gonadectomy) at <6 months, 6 to 12 months, >12 months or entire. Surgical method and number of dogs in each neuter group not reported | DOD (first occurrence of veterinarian-reported CCL rupture or osteoarthritis) Obesity (veterinarian-reported overweight or obese) | Annual questionnaires from veterinarians and medical records Overweight / obesity–Purina Body Condition Score scale of >6/9 | Retrospectively grouped | Between age group comparisons reported |
| Sonnenschein et al., 1991 [21] | USA | CC | 428 female (150 cases, 147 cancer controls, 131 non-cancer controls) | Examine relationships between dietary intake and body conformation and the risk of breast cancer | Neutering, surgical method not stated | Neoplasia (mammary) | Veterinary records | Retrospectively grouped | No comparisons reported between dogs neutered at different ages |
| Spain et al., 2004 [12] | USA | RC | 1,659 (number of females not reported) | Evaluate the long-term risks and benefits of early-age (before 5.5 months) versus traditional-age gonadectomy | Neutered <5.5 months or ≥5.5 months, number breakdown for each neuter age group not available. Surgical method not stated | DOD (CCL, HD, patella luxation) Neoplasia Obesity (overweight body condition) Urogenital (perivulvar dermatitis, UI, urolithiasis, UTI-cystitis, vaginitis) | HD, UI, neoplasia, vaginitis–owner questionnaire and veterinary records, CCL, patella luxation, perivulvar dermatitis, UTI-cystitis, urolithiasis–veterinary records Overweight body condition–owner reported body shape determined by comparison to 5 images | Retrospectively grouped | Between age group comparisons reported |

(Continued)

**Table 1.** (Continued)

| Author and year | Source / country | Study design | Sample size | Aims* | Intervention | Relevant health outcomes* | Outcome measures | Allocated randomly to neutering groups or retrospectively grouped | Comparisons reported between dogs neutered in different groups or just to the entire group? |
|---|---|---|---|---|---|---|---|---|---|
| Stocklin-Gautschi *et al.*, 2001 [56] | Switzerland | RC | 206 female | Determine the influence of early neuter on the incidence of UI in bitches | OVH or ovariectomy, all before puberty | Urogenital (UI) | Not stated | Retrospectively grouped | Before and after puberty compared |
| Sundberg *et al.*, 2016 [57] | USA | RC | 6,281 dogs with age at neuter data (number of females not reported) | Examine associations between neuter status and diseases thought to be modulated by the immune system | Neutering, surgical method not stated | Atopy | Veterinary records | Retrospectively grouped | No comparisons reported between dogs neutered at different ages |
| Thrusfield *et al.*, 1998 [24] | UK | PC | 310 female | Determine the incidence of AUI in neutered and entire bitches and assess the effects of time of neutering | Neutering, surgical method not stated; 92 neutered before first oestrus, 218 after first oestrus | Urogenital (AUI) | Veterinarian questionnaires | Based on owner decision | Before and after puberty compared |
| Torres de la Riva *et al.*, 2013 [13] | USA | RC | Males and females; 364 females (242 neutered) | Examine the effects of neutering on disease risk distinguishing between early or late neutering compared to remaining entire | Neutering, surgical method not stated; 172 females neutered <12m and 70 females neutered ≥12m | DOD (HD, ED, CCL) Neoplasia (LSA, HSA, MCT, MC, OSA) | Veterinary records | Retrospectively grouped | Between age group comparisons reported |
| Veronesi *et al.*, 2009 [58] | Italy | Unclear | 750 female | Examine the influence of age at neutering on the occurrence of UI | OVH or ovariectomy | Urogenital (UI) | Veterinary records | Retrospectively grouped | Relationships with age at neuter reported |
| Whitehair *et al.*, 1993 [59] | North America | RC | 10,769 dogs. Age at neuter known for 244. | Determine the prevalence of CCL rupture in dogs and whether CCL rupture was associated with age, breed, gender or bodyweight | OVH, method of grouping by neuter age not described | DOD (CCL rupture) | Veterinary records | Unclear | Relationships with age at neuter reported |

(*Continued*)

**Table 1.** (Continued)

| Author and year | Source / country | Study design | Sample size | Aims* | Intervention | Relevant health outcomes* | Outcome measures | Allocated randomly to neutering groups or retrospectively grouped | Comparisons reported between dogs neutered in different groups or just to the entire group? |
|---|---|---|---|---|---|---|---|---|---|
| Zink *et al.*, 2014 [60] | 25 different countries, majority USA, UK, Canada and Australia (USA 86.7%) | CS | 2,505 (1,360 female) | Investigate associations between age at gonadectomy and sex of gonadectomised dogs and risk and age of onset of neoplasia | Owner selected age at gonadectomy: ≤6 months (209 female), 7–12 months (157 female) and >12 months (459 female), or entire. Surgical method not stated | Neoplasia (HSA, LSA, MCT, all other cancers, and all cancers combined) | Owner survey | Retrospectively grouped | No comparisons reported between dogs neutered at different ages |
| Zlotnick *et al.*, 2019 [61] | USA | RC | 245 (77 female) | Investigate differences in the incidence of health or behavioural problems and completion of service dog training in dogs neutered at different ages | Neutered at <7 months (n = 110) Neutered at 7–11 months (n = 58) Neutered at > 11 months (n = 77). Surgical method not stated | DOD (dismissal from service dog programme for orthopaedic reasons) | From service dog records | Retrospectively grouped | Between age group comparisons reported |

*For the purposes of this scoping review, only health aims and outcomes relevant to the subject of the review are included. Other aims or outcomes which may have been included in the studies, such as other health outcomes, or behaviour are not reported.

**Abbreviations:** RC–retrospective cohort, PC–prospective cohort, CS–cross-sectional, CC–case-control, AUI–acquired urinary incontinence, CCL–cranial cruciate ligament, DOD–developmental orthopaedic disease, ED–elbow dysplasia, OVH–ovariohysterectomy, HD—hip dysplasia, HSA–haemangiosarcoma, LSA–lymphosarcoma/lymphangiosarcoma, MC–mammary cancer, TPA–tibial plateau angle, UI—urinary incontinence, USMI–urinary sphincter mechanism incompetence, UTI–urinary tract infection.

studies reported no impact on urogenital disease (acquired sphincter mechanism incompetence [25]; acquired UI [24, 26]; problems associated with the urinary system [11], UI [54]). One study did not analyse data statistically due to small numbers of dogs [46]. Seven studies did not compare between neuter groups; the only comparisons made were between each neuter group and entire dogs [14–17, 21, 55, 60] and one study only compared between ages for one outcome measure, neoplasia and not the second, UI [45]. These studies where no effects were identified or that did not compare between different neuter groups were not included in the charting table.

One study [54] aimed to investigate the impacts of neutering before or after the first oestrus, however data availability was too limited to enable analysis. Additionally, this study only reported comparisons for all neuter age groups to bitches neutered at 6 to <12 months; no comparisons were reported between other neuter age groups. However, the author has confirmed that these comparisons were made and there were no differences identified (C. Pegram, *Pers. Comm.*).

**Table 2. Study population characteristics for the publications identified for inclusion in the scoping review of the literature on the effect of the timing of neutering in relation to puberty on atopy, developmental orthopaedic disease (DOD), neoplasia, obesity and urogenital disease in female domesticated dogs.**

| Author and year | Sample source | Breed | Study duration | Sex | Consider health before neuter | Excluded dogs with a related health problem before neuter? | Previous breeding history considered? | How puberty was defined |
|---|---|---|---|---|---|---|---|---|
| Beaudu-Lange et al., 2021 [45] | Pet dogs identified from veterinary clinic records | Various breeds and cross breeds | To at least 6 years of age, to death for 293 bitches | All female | Yes | No | No | N/A, puberty not considered within the study, only age |
| Brandli et al., 2021 [46] | Pet dogs identified from clinical records from the Clinic for Small Animals, University of Zurich | 31 purebred (14 different breeds) one mixed-breed | Varied from 0.7 to 11.3 years. Mean observation period 5.3 ± 3.4 years. | All female | No | No | Yes–all nulliparous | Not reported |
| Byron et al., 2017 [47] | Not stated, data from veterinary practices | Various breeds, not stated | N/A–CC design | All female | No | No | No | N/A, puberty not considered within the study, only age |
| Cooley et al., 2002 [48] | Pet dogs recruited via breed clubs and advertising | Rottweilers | Varied from 1.3 to 15.6 years. Mean observation period 8.8 and 8.6 years for cases and controls respectively | Both male and female, reported separately | Yes | Unclear | Yes for entire and neutered >1 year | N/A, puberty not considered within the study, only age |
| de Bleser et al., 2011 [25] | Not stated, data from veterinary practices | Various breeds and cross breeds | N/A–CC design | All female | Yes | Yes | No | Before or after first oestrus |
| Duerr et al., 2007 [49] | Not stated, data from veterinary surgeons and university veterinary medical centre records | Various breeds and mixed breeds | N/A–CC design | Both male and female, results not reported by sex | No | No | No | N/A, puberty not considered within the study, only age |
| Ekenstedt et al., 2017 [19] | Mainly pet dogs from referral hospitals | Labradors | N/A–CC design | Both male and female, reported separately | No | Yes for controls | No | N/A, puberty not considered within the study, only age |
| Forsee et al., 2013 [26] | Not stated, data from veterinary practices and hospitals | Breeds not stated | 4 to 7 years post-neuter | All female | Yes | Yes | Yes | N/A, puberty not considered; whether oestrus cycles occurred before OVH was asked, but not reported |
| Hart et al., 2014 [14] | Not stated, data from veterinary hospital | Labradors and Golden Retrievers | To a maximum of 8 years of age for individual dogs | Both male and female, reported separately | Yes | No but dogs with health problem before neuter were classed as entire for that disease | No | N/A, puberty not considered within the study, only age |

(*Continued*)

**Table 2.** (Continued)

| Author and year | Sample source | Breed | Study duration | Sex | Consider health before neuter | Excluded dogs with a related health problem before neuter? | Previous breeding history considered? | How puberty was defined |
|---|---|---|---|---|---|---|---|---|
| Hart *et al.*, 2016 [15] | Not stated, data from veterinary hospital | German Shepherd Dogs | To a maximum of 8 years of age for individual dogs | Both male and female, reported separately | Yes | No but dogs with health problem before neuter were classed as entire for that disease | No | N/A, puberty not considered within the study, only age |
| Hart *et al.*, 2020a [16] | Not stated, data from veterinary hospital | 35 different pure breeds | To a maximum of 11 years of age for individual dogs 15 years of data for most breeds | Both male and female, reported separately | Yes | No but dogs with health problem before neuter were classed as entire for that disease | No | N/A, puberty not considered within the study, only age |
| Hart *et al.*, 2020b [17] | Not stated, data from veterinary hospital | Cross breeds | To a maximum of 11 years of age for individual dogs | Both male and female, reported separately | Yes | No but dogs with health problem before neuter were classed as entire for that disease | No | N/A, puberty not considered within the study, only age |
| Howe *et al.*, 2001 [11] | Pet dogs adopted from two shelters | Breeds not stated | 41 to 64 months after neutering surgery | Both male and female, results not reported by sex | No | No | No | Based on age at neutering, ≥24 or <24 weeks of age |
| Lefebvre *et al.*, 2013 [50] | Pet dogs | Various breeds and cross breeds | 10 years or more of regular veterinary hospital visits | Both male and female, results not reported by sex | Yes | Yes | No | N/A, puberty not considered within the study, only age |
| Lutz *et al.*, 2019 [22] | Not stated, data from two veterinary hospitals | Various, not stated | 4.9 to 15.6 years after neuter | All female | Not stated | Not stated | No | Not defined, however some dogs neutered before puberty were max 1.4 years of age and some neutered after puberty were 0.3 years of age so unreliable classification |
| Moxon *et al.*, 2023 [51] | Guide Dogs UK | Labrador and Golden Retriever cross breeds | To 17 months of age | All female | Yes | No | Yes–all nulliparous | Before or after first oestrus |
| Palerme *et al.*, 2021 [52] | Not stated, data from specialist referral centre | Various, not stated | N/A–CS design | All female | No (those already received vulvoplasty were excluded) | No | No | N/A, puberty not considered within the study, only age although suggest pre- or peri-pubertal neutering prior to 4 or 8 months of age |
| Pegram *et al.*, 2019a [53] | Not stated, data from primary-care veterinary practices | Various breeds and cross breeds | Up to 8 years | All female | Yes | Yes | No | N/A, puberty not considered within the study, only age |

(*Continued*)

**Table 2.** (Continued)

| Author and year | Sample source | Breed | Study duration | Sex | Consider health before neuter | Excluded dogs with a related health problem before neuter? | Previous breeding history considered? | How puberty was defined |
|---|---|---|---|---|---|---|---|---|
| Pegram *et al*., 2019b [54] | Not stated, data from primary-care veterinary practices | Various breeds and cross breeds | N/A–CC design | All female | Yes | Yes | No | Before or after first oestrus |
| Salmeri *et al*., 1991 [10] | Dogs in a research kennel facility | Mixed-breed | Dogs followed from 0 to 15 months of age | Both male and female, reported separately | Yes | No | No | N/A, puberty was not considered within the study, only age |
| Schneider *et al*., 1969 [20] | Pet dogs identified from neoplasm registry | Various breeds and cross breeds | N/A–CC design | All female | Yes | No | Yes–not controlled for in oestrous cycle analysis | Based on number of oestrous cycles before neutering |
| Simpson, 2017 [55] | Pet, data from the Golden Retriever Lifetime Study | Golden Retrievers | Not stated | Both male and female, results not reported by sex | Not stated | Not stated | No | N/A, puberty was not considered within the study, only age |
| Simpson *et al*., 2019 [18] | Pet, data from the Golden Retriever Lifetime Study | Golden Retrievers | Not stated | Both male and female, results not reported by sex for age at neutering | Yes | Yes | No | N/A, puberty was not considered, only age, although suggest <6 months–prepubertal; >6 to <12 months–dogs in puberty; >12 months post-pubertal |
| Sonnenschein *et al*., 1991 [21] | Pet, data from School of Veterinary Medicine, University of Pennsylvania | Various breeds and cross breeds | N/A–CC design | All female | Yes | No, unclear whether dogs diagnosed before neutering were still included | Yes | N/A, puberty was not considered within the study, only age |
| Spain *et al*., 2004 [12] | Pet dogs adopted from a shelter | Various breeds, not stated | Median 4.5 years (range 0.3 to 11.3 years) | Male and female, sex ratio in each age at neuter group not reported for overweight analysis. Results reported separately for UI and cystitis | No | No | No (but maximum age at neuter was 12 months so breeding unlikely) | N/A, puberty was not considered within the study, only age |
| Stocklin-Gautschi *et al*., 2001 [56] | Not stated | Not stated | A minimum of 3 years post-neuter | All female | Yes | Yes | No | Data for bitches neutered before puberty were compared to those for bitches neutered after puberty from a different study |
| Sundberg *et al*., 2016 [57] | Pet dogs, data from a veterinary medical teaching hospital | Various breeds and cross breeds | Not stated | Both male and female, results not reported by sex for age at neutering | Yes | No | No | N/A, puberty was not considered within the study, only age |

(*Continued*)

**Table 2.** (Continued)

| Author and year | Sample source | Breed | Study duration | Sex | Consider health before neuter | Excluded dogs with a related health problem before neuter? | Previous breeding history considered? | How puberty was defined |
|---|---|---|---|---|---|---|---|---|
| Thrusfield et al., 1998 [24] | Pet dogs, data from veterinary practices | Various breeds and cross breeds | Five years | All female | Not stated, but all UI occurred after neutering | Yes | Yes | Before or after first oestrus |
| Torres de la Riva et al., 2013 [13] | Not stated, data from veterinary hospital | Golden Retriever | To a maximum of 8 years of age for individual dogs | Both male and female, reported separately | Yes | No but dogs with health problem before neuter were classed as entire for that disease | No | N/A, puberty was not considered within the study, only age |
| Veronesi et al., 2009 [58] | Pet dogs, source not stated | Various breeds and cross breeds | Not clear | All female | Not stated, but all UI occurred after neutering | No | No | N/A, puberty was not considered within the study, only age |
| Whitehair et al., 1993 [59] | Pet dogs, veterinary medical teaching hospital records | Various breeds and cross breeds | Not clear | Both male and female, results not reported by sex for age at neutering | No | No | No | N/A, puberty was not considered within the study, only age |
| Zink et al., 2014 [60] | Pet dog data gathered from an anonymous online survey | Viszlas | Up to 16 years | Both male and female, results not reported by sex for age at neutering | No | No | Yes for entire (53%), not mentioned for neutered | N/A, puberty was not considered within the study, only age |
| Zlotnick et al., 2019 [61] | Dogs in a service dog programme | Labradors, golden retrievers, various other breeds and crosses | Not stated | Both male and female. Sex breakdown by neuter group not reported, results not reported by sex | Yes | No | No | N/A, puberty was not considered within the study, only age |

**Abbreviations:** CS–cross-sectional, CC–case-control, UI—urinary incontinence, N/A–not applicable

The effects of neutering at younger and older ages were mixed, with earlier neutering commonly reported to be detrimental for DOD and urogenital disease, but beneficial in reducing the incidence or risk of neoplasia. For some diseases, findings were also mixed; four of the six studies that reported significant findings for UI found earlier neutering to be detrimental, whilst two suggested earlier neutering was favourable. However, one of the latter studies was primarily investigating onset of UI post-neuter, rather than focusing on comparisons of UI incidence in bitches neutered at different ages. The other study made comparisons between bitches in the study that were neutered before puberty and bitches from another study that were neutered post-pubertally. Four of the five studies that reported significant findings for neoplasia found earlier age at neuter to be favourable, however, one study examining the risk of bone sarcoma found an increased risk with fewer months entire (therefore earlier age at neuter). Spain et al. [12] reported mixed effects of neutering age on HD, with increased hazard ratio in dogs neutered younger, but increased risk of being euthanised for HD for dogs with HD that had been neutered at older ages.

The charting of study results highlights the inconsistencies in classification of timing of neutering between studies and the lack of information concerning the timing of neutering in relation to puberty.

**Table 3. The health outcomes that were impacted by timing of neutering identified in a scoping review of the literature on the effect of the timing of neutering in relation to puberty on atopy, developmental orthopaedic disease (DOD), neoplasia, obesity and urogenital disease in female domesticated dogs.**

| Health outcome | Author | Study design | Sex | Relationship with timing of neutering | Age or puberty | Summary impact of neutering for each health outcome studied |
|---|---|---|---|---|---|---|
| Developmental orthopaedic disease | Duerr *et al.*, 2007 [49] | CC | Both male and female | Dogs with TPA ≥35˚ were 3 times (95% CI 1.2–8.0) as likely to have been neutered <6 months of age than dogs with TPA <35˚.<br>Dogs with TPA ≥35˚ in both limbs were 13.6 times (95% CI 2.72–68.1) as likely to have been neutered <6 months of age as were dogs with TPA ≤30˚ in both limbs. | Age | Less favourable for dogs neutered at <6 months of age |
| | Ekenstedt *et al.*, 2017 [19] | CC | Female | Increased odds of CCL rupture for bitches neutered ≤1 year of age compared to those neutered >1 year of age (OR = 4.30, 95% CI 1.55–12.72, P = 0.0021). | Age | Less favourable for bitches neutered at ≤1 year of age |
| | Simpson *et al.*, 2019 [18] | PC | Both male and female | Increased risk of orthopaedic injury in dogs neutered ≤6 months compared to 6 to ≤12 months (HR = 2.24, 95% CI–1.25–4.93, P = 0.008) and to >12 months (HR = 6.36, 95% CI 2.74–14.77, P<0.0001).<br>Increased risk of orthopaedic injury in dogs neutered 6 to ≤12 months compared to >12 months (HR = 2.56, 95% CI–1.08–2.25, P = 0.03).<br>No influence of sex on the association between neutering and orthopaedic injuries (P = 0.23). Results for each sex not presented separately. | Age | Less favourable for dogs neutered at younger ages |
| | Spain *et al.* (2004) [12] | RC | Both male and female | Higher HR for HD in dogs neutered <5.5m than those neutered ≥5.5 months (HR = 1.7, 95% CI 1.04–2.78, P = 0.03) due to earlier age at diagnosis in dogs neutered <5.5m.<br>Dogs with HD that were neutered ≥5.5 months of age were 3x as likely to be euthanised for HD as those with HD that were neutered <5.5 months of age (P = 0.02). | Age | Less favourable for dogs neutered <5.5 months of age<br>More favourable for dogs neutered <5.5 months of age |
| | Torres de la Riva *et al.*, 2013 [13] | RC | Female | Occurrence of CCL tear increased for bitches neutered <12 months (7.7%) compared to those neutered ≥12 months of age (0.0%) (P = 0.001). | Age | Less favourable for bitches neutered <12 months compared to ≥12 months |
| | Zlotnick *et al.*, 2019 [61] | RC | Both male and female | Dogs neutered <7 months over 2x as likely to be dismissed from the service dog training programme for orthopaedic problems as dogs neutered at any older age (RR = 2.2, 95% CI 1.1–4.3, P = 0.03). Results for females not reported separately but stated that health-related dismissals did not differ by sex or breed. This is dismissals *not* diagnoses. | Age | Less favourable for dogs neutered <7 months |
| Neoplasia | Beaudu-Lange *et al.*, 2021 [45] | RC | Female | Early neutered (<2 years of age) females had a significantly lower risk of developing mammary tumours than late neutered or entire bitches (OR = 0.10, 95% CI 0.01–0.41, P<0.001). | Age | More favourable for earlier neuter |
| | Cooley *et al.*, 2002 [48] | RC | Female | Females that developed bone sarcoma were entire for fewer months than those that did not (HR = 0.98, 95% CI 0.97–0.99, P<0.0001). For each additional month of being sexually entire, there was a 1.4% reduction in bone sarcoma risk.<br>Negative association between gonadal hormone exposure and risk of bone sarcoma (P = 0.006 for females). In females neutered <1 year of age bone sarcoma incidence rate was >3x greater than for entire females (RR = 3.1, 95% CI 1.1–8.3, P = 0.02). There was no difference between entire bitches and those neutered >1 year of age. | Age | More favourable as age at neuter i.e. months remained entire increases |
| | Schneider *et al.*, 1969 [20] | CC | Female | Neutered bitches had lower risk of mammary neoplasia than entire multi-oestrus bitches, this was significant for those neutered before the first oestrus (RR = 0.005). | Puberty | More favourable for bitches neutered before puberty |
| | Torres de la Riva *et al.*, 2013 [13] | RC | Female | Bitches neutered ≥12 months of age more likely to have HSA (RR = 7.48, 95% CI 1.79–31.30) and MCT (RR = 4.46, 95% CI 1.11–17.82) than those neutered <12months of age. | Age | More favourable for bitches neutered earlier |

*(Continued)*

**Table 3.** (Continued)

| Health outcome | Author | Study design | Sex | Relationship with timing of neutering | Age or puberty | Summary impact of neutering for each health outcome studied |
|---|---|---|---|---|---|---|
| Obesity | Simpson et al., 2019 [18] | PC | Both male and female | Dogs neutered 6–12 months had an increased risk for overweight/obesity compared to those neutered >12 months of age (HR = 1.42, 95% CI 1.19–1.69, P = 0.0001). Dogs neutered <6 months not different to dogs neutered 6–12 months (HR = 0.82, 95% CI 0.66–1.01, P = 0.06) or >12 months (HR = 1.16, 95% CI 0.93–1.44, P = 0.19). | Age | Less favourable for dogs neutered at 6 to 12 months compared to those neutered at >12 months of age |
| | Spain et al., 2004 [12] | RC | Both male and female | The prevalence of overweight body condition declined with decreasing age at neuter (OR for each 1-month decrease in age at neuter 0.94; P = 0.04). | Age | More favourable for dogs neutered younger |
| Urogenital disease | Byron et al., 2017 [47] | CC | Female | No significant difference in median age at OVH in isolation between UI and control dogs (both 8.5 months). An increased hazard of UI with higher adult body weight and earlier OVH was found that was statistically significant at 25 kg body weight (HR = 0.89, 95% CI 0.82–0.97, P = 0.006). For a 30 kg dog, a 1-month increase in OVH age decreased the hazard of UI by 24% (HR = 0.86, 95% CI 0.78–0.94, P = 0.001). | Age | More favourable as age at neuter increases–but not at every bodyweight |
| | Lutz et al., 2019 [22] | RC | Female | UI reported in fewer dogs neutered after puberty (10.7%) than before puberty (22.9%; P = 0.007). No OR or CI reported. | Puberty | Less favourable for bitches neutered before puberty |
| | Moxon et al., 2023 [51] | PC | Female | Significantly more bitches neutered prepubertally had vulvas that changed to be smaller by 17 months of age (length: P<0.001; width: P<0.001). Bitches neutered prepubertally had smaller changes in vulval length (0.08 ± 0.06 cm vs 0.41 ± 0.05 cm, P<0.001) and width (-0.05 ± 0.04 cm vs 0.20 ± 0.04 cm, P<0.001) between 6 and 17 months of age. Significantly more prepubertally neutered bitches had vulvas that appeared juvenile (P<0.001) and recessed (P = 0.005) in appearance on direct observation and more categorised as recessed / inverted on examination of digital images (P = 0.002). | Puberty | Less favourable for bitches neutered before puberty |
| | Palerme et al., 2021 [52] | CS | Female | Bitches neutered ≤12 months of age had significantly increased OR of recessed vulva than those neutered >12 months (OR = 2.95, 95% CI 1.24–7.03, P = 0.015. Bitches neutered ≤24 months of age had significantly increased OR of recessed vulva than those neutered >24 months (OR = 3.08, 95% CI 1.48–8.26, P = 0.026). No difference between ≤4m vs >4m or ≤8m vs >8m. | Age | Less favourable for bitches neutered earlier |
| | Pegram et al., 2019a [53] | RC | Female | Bitches neutered <6 months of age had significantly increased hazard of early-onset UI within the first year after neuter (HR = 1.82; 95% CI 1.15–2.88; P = 0.011). The HR decreased 0.75-fold for every subsequent year. | Age | More favourable as neutering age increases |
| | Spain et al., 2004 [12] | RC | Female | Incidence of cystitis higher for females neutered <5.5 months of age (HR = 2.76, 95% CI 1.08–7.14, P = 0.02). Decreasing age at neuter associated with increasing incidence of UI (HR for each 1-month decrease in age at neuter 1.2; 95% CI 1.06–1.35, P<0.01). Females neutered <3 months of age at highest risk, compared to those neutered ≥3 months (HR = 3.46; P<0.001). | Age | More favourable as neutering age increases |
| | Stocklin-Gautschi et al., 2001 [56] | RC | Female | Incidence of UI in bitches neutered prepubertally (9.7%) was significantly lower than the incidence for bitches neutered post-pubertally in a different study (20.1%; P<0.05). | Puberty | More favourable for bitches neutered before puberty |
| | Veronesi et al., 2009 [58] | Unclear | Female | UI occurred earlier in bitches neutered at older ages. The authors suggest that this confirms a lower risk in bitches neutered at earlier ages. | Age | Favourable for bitches neutered earlier |

**Abbreviations:** RC–retrospective cohort, PC–prospective cohort, CS–cross-sectional, CC–case-control, CI–confidence interval, HR–hazard ratio, OR–odds ratio, RR–relative risk, CCL–cranial cruciate ligament, OVH–ovariohysterectomy, HD—hip dysplasia, TPA—tibial plateau angle, UI–urinary incontinence.

## Discussion

This is the first scoping review that collates and describes the literature relating to the effects of neutering bitches before or after puberty on several health outcomes. The review identified a lack of evidence: 33 studies were eligible for inclusion, but only six categorised the bitches as being surgically neutered either before or after puberty. One study considered bitches that were pre or post-pubertal at the time of the first treatment with deslorelin acetate for oestrus suppression and 26 examined the effects on health related to age, rather than pubertal status, at neutering.

This scoping review suggests that robust evidence to support veterinarians, those working with dogs and pet dog owners when discussing the timing of neutering relative to puberty does not yet exist. The impact of neutering *before or after puberty* on atopy, DOD, neoplasia, obesity and urogenital disease in female domesticated dogs remains unknown. This information is important for veterinarians to support conversations around the health impacts of neutering in relation to puberty. This study summarises the current state of the evidence and provides direction for future studies.

Five of the six studies that considered neutering before or after puberty involved various breeds, included between 32 and 370 bitches and examined urogenital disease (five studies), or mammary neoplasia (one study). For one of these studies involving 101 matched pairs [22], method of determining pubertal status was not defined and potential problems with categorising bitches as pre/post-pubertal at neuter as evidenced by the ages at neutering of the bitches may reduce confidence in the results. Within the study that considered bitches that were pre or post-pubertal at the time of the first treatment with deslorelin acetate for oestrus suppression, only four bitches were treated before puberty [46]. Stocklin-Gautschi *et al.* [56] compared data to those from a group of bitches from another similar study to draw conclusions about the differences in incidence of UI for bitches neutered before or after the first oestrus. The study was excluded from a previous systematic literature review due to risk of bias. Additionally, the source of data regarding UI diagnosis was not stated. The studies by Schneider *et al.* [20] and Thrusfield *et al.* [24] have previously been reviewed and found to be compromised by a lack of controlling for confounders and problems with participant recruitment [1, 2]. For three of the four studies examining UI or USMI, there is a risk of bias due to unconfirmed diagnoses using owner-reported data.

The remaining 26 studies reported data for dogs based on age at neutering and considered or grouped age in different ways (see Table 1). This does not allow for easy between-study comparisons to be made, nor does it allow conclusions to be drawn about the impact of neutering in relation to puberty. One of the studies that grouped dogs as neutered at either less than 24 weeks of age, or 24 weeks of age or more, reported in the abstract that the results indicate that neutering before puberty is not associated with increased incidence of physical problems. Additionally, the term 'prepubertal' was used throughout the discussion to demonstrate no differences in health problems between those that underwent 'prepubertal' neutering compared to 'traditional age' neutering. However, this is misleading based on the age grouping of dogs at the time of neutering. Breeding history and previous pregnancies were commonly not included in studies, and may have impacted the findings.

Nine of the 18 studies that included male and female dogs did not report results by sex separately. Differences in incidence of certain diseases between male and female dogs have previously been reported [16, 62] and this prevents a clear understanding of the impacts specifically on bitch health. Additionally, 25 studies did not consider previous breeding history, which may have affected disease occurrence.

Most studies included various or multiple breeds and did not state the breeds involved, or present results by breed, however some studies dealt with this by controlling for breed in the

analyses or by matching control group bitches by breed. Breed can impact disease incidence, with some breeds predisposed to certain diseases such as joint disorders, neoplasia and UI [14, 16, 17, 23]. Additionally, breed can impact age at puberty, with smaller breeds that reach adult bodyweight earlier experiencing puberty sooner [37]. Therefore, understanding the breeds, and controlling for breed in analysis is important so that results are not influenced by the inclusion of high or low numbers of certain breeds. However, this does pose difficulties for future study design. Undertaking individual studies across different geographical locations–to account for variations that may be present in different subpopulations—for many breeds, or recruiting a sufficient number of bitches of each breed to one prospective study would not be easy.

Studies also had different follow up times, with some not considering diseases that affected dogs in older age. While some authors stated a wish to reduce the impact of old age on disease occurrence to allow the focus to remain on the effect of neutering, dogs with some diseases such as HD or mammary neoplasia may not be expected to present with clinical signs until later in life [63–67]. There were differences between studies, and in some cases between dogs within studies, in the time elapsed since neutering, and in the length of time dogs were followed for, which was not always described or controlled for. These are important considerations because such factors may not have allowed time for certain diseases to be observed.

Fifteen studies did not report whether they excluded dogs that were already diagnosed with the disease prior to neutering, and for a further four studies this information was not clearly stated. When measuring the impact of an intervention on an outcome, it is important to either understand the status of the outcome prior to the intervention or to exclude those already affected from the study. For instance, if a study was examining the impact of a treatment or intervention on the occurrence of a disease, prior to administration, study participants would all need to have been free from the disease. If they were not, or their disease status was unknown, understanding of the impact could be compromised by the fact that some participants may have been affected throughout the study, and to differing degrees. Similarly, if the dogs were already diagnosed prior to being neutered, neutering itself could not be deemed a risk factor for that disease being present in the follow up period post-neuter. The lack of consideration for dogs' relevant disease status prior to neutering, or lack of exclusion of dogs already diagnosed prior to neutering, impacts the confidence in the results.

## Summary of the evidence

Despite the lack of evidence relating specifically to puberty, this scoping review provided an overview of the available literature on the effects of neutering timing on health in female dogs. Only seven studies were identified that categorised bitches as pre or post-pubertal at surgical neutering or first treatment for oestrus suppression, highlighting a need for further research and the importance of reporting this critical factor in studies. None of these studies examined atopy, DOD, or obesity, one considered mammary neoplasia and five investigated outcomes relating to urogenital disease.

**Atopy.** Only one study was identified that investigated the effect of neutering timing in relation to atopy [57]. The study considered age and not pubertal status at neutering, and although the authors suggested that age at neutering was to be examined, disease rates for age at neutering were not reported. The authors did report no significant differences in "age at gonadectomy as a function of disease (P>0.1)". However, which diseases were analysed and disease incidence data by age at neuter were not reported.

**Developmental orthopaedic disease.** All of the 13 studies that investigated DOD considered timing of neutering in relation to age rather than pubertal status. Five did not compare

disease incidence between neuter age groups, only between each neuter age neuter group and a group of entire dogs [14–17, 55]. Eight studies did make comparisons, two of those reported no impact of neutering age on DOD [11, 59].

Six studies reported effects of neutering age on DOD, specifically on CCL rupture [13, 19], tibial plateau angle in dogs with CCL [49], HD [12], "orthopaedic injury" [18] and dismissal from a service dog programme for orthopaedic problems [62]. Despite differences between studies in how age at neuter was grouped, in all six studies younger age at neuter groups were at increased risk of the particular outcome. In one study [12], secondary to the main outcome of incidence of HD, dogs that had HD and were neutered at ≥5.5 months of age were more likely to be euthanised for the disease than those dogs with HD that were neutered before 5.5 months of age. Only two of the studies [13, 19] reported results for female dogs separately.

**Neoplasia.** Only one of the 11 studies that investigated neoplasia considered pubertal status at neutering [20]. In that case-control study malignant mammary neoplasia was less likely in bitches neutered before experiencing any oestrus (relative risk 0.005) compared to those that had experienced one oestrus before neutering (relative risk 0.08) and two or more oestrous cycles before neutering (relative risk 0.26). However, as reported by Beauvais et al. [1], there is a risk of bias due to recruitment of cases and controls from different time periods and controlling for confounding variables in analysis.

The remaining 10 studies investigated the effects of timing of neutering related to age. Six of these made no comparisons of disease incidence between age at neuter groups, only between each age at neuter group and a group of entire dogs [14–17, 21, 60]. Four studies did make comparisons and one of those reported no impact of neutering age [12].

Three studies did report effects of neutering age, specifically on mammary tumours [45]; appendicular bone sarcoma [48] and HSA and MCT [13]. Beaudu-Lange et al. [45] reported a protective effect of neutering before two years of age on mammary tumour development. While this is similar to the findings of Schneider et al. [20], the study included a relatively small number of bitches (n = 50) that were neutered <2 years of age compared to those neutered >2 years (n = 243). Forty-seven of the bitches neutered >2 years of age were known to have a mammary tumour at the time of neutering and do not appear to have been excluded from the analysis. Torres de la Riva et al. [13] also reported more favourable outcomes for earlier neutering for HSA and MCT for 242 female Golden Retrievers. In that study, bitches were grouped as those neutered before (n = 172) or at/over 12 months of age (n = 70). Data were gathered from veterinary hospital records which could affect how well they represent the general dog population, however, confirmation of diagnosis was reliable. Bitches were studied to nine years of age, which could impact disease occurrence reporting. Additionally, there was no difference between bitches neutered before or at/after 12 months of age in the incidence of LSA although, in contrast to other types of neoplasia, a higher percentage of bitches in the early (5.9%) rather than late neuter group (1.4%) were diagnosed. Bitches with disease prior to neutering were excluded, the study only included one breed and sex, addressing the potential for these confounding variables to impact the results.

In contrast to the first two studies, Cooley et al. [48] found an increased risk with younger age at neuter for bone sarcoma in 683 Rottweilers, 389 of which were female. Data for neuter age were considered as months sexually entire on a continuous scale and were also grouped into neuter age groups and were reported for males and females separately. While data were gathered from an owner questionnaire, completed with veterinarian assistance, diagnoses were confirmed by reviewing radiographs which reduced the risk of incorrect diagnoses. Data on dog height and weight were included as confounding variables. It is not clear whether there were any dogs with bone sarcoma diagnosed before neutering and no mention of whether such cases were excluded from analysis.

**Obesity.** None of the six studies that investigated obesity considered timing of neutering by pubertal status; all investigated effects of neutering timing related to age. One of the studies did not compare disease incidence between neuter age groups, only between each neuter age group and a group of entire dogs [55]. Five studies did make comparisons and three of those reported no impact of neutering age on overweight/obesity [10, 11, 50].

Two studies reported effects of age at neutering on obesity with contrasting results [12, 18]. Both studies considered being overweight and obese rather than obesity alone. Spain et al. [12] reported a decreasing risk of overweight body condition with decreasing age at neuter for 1,659 dogs of various breeds, although results were not reported separately for male and female dogs. Data validity may have been compromised due to owners reporting overweight/obese status on a questionnaire by comparing their dog's body shape to five images, and because age was estimated in some cases for the shelter dogs. Confounding variables were reportedly controlled for, although it is unclear which variables specifically were included in the analyses. Simpson et al. [18] examined data for over 2,700 Golden Retrievers, however the exact number of dogs, and the number neutered in each age group was not clear. The results suggested no difference in age at neuter between those neutered at <6 and those neutered at 6–12 or >12 months of age. However, dogs neutered at 6–12 months had an increased risk for overweight/obesity compared to those neutered at >12 months of age. Dogs that were overweight prior to neutering were excluded, data were reported by veterinarians on an annual questionnaire and were based on a body condition scoring system, and confounding variables such as sex, physical activity level and age were controlled for.

**Urogenital disease.** Six of the 18 studies that investigated urogenital disease (four UI, one USMI, one vulval development) considered pubertal status at the time of neutering [22, 24, 25, 46, 51, 56]. One of these [46] did not have a large enough sample size to enable statistical comparisons, and two reported no significant effects [24, 25]. The strengths and risk of bias for these two studies have previously been reported by Beauvais et al. [2]. Three studies did report effects of neutering before or after puberty on urogenital disease [22, 51, 56]. Stocklin-Gautschi et al. [56] suggested that the incidence of UI in bitches neutered prepubertally was lower than for bitches neutered post-pubertally, however there were issues with study methodology and a risk of bias that may compromise the results. The study only included bitches neutered before puberty and compared data to those for bitches that were neutered after puberty in a different study. The breeds and source of the bitches included and how outcomes were measured were not stated and the controlling for effects of confounding variables was not clear [2]. Lutz et al. [22] reported the opposite, with significantly more bitches neutered before puberty being reported with UI than those neutered after puberty. However, whether bitches with UI before neutering were excluded was not stated, UI was reported on an owner questionnaire and no OR or CI were reported. Importantly, results may be compromised by the grouping of bitches. There were some bitches aged up to 1.4 years in the group neutered before puberty and some aged as young as 0.3 years in the group neutered after puberty (presumably these are data errors).

The final study reported effects on vulval size and appearance for 306 Labrador / Golden Retriever crossbreed bitches [51]. Significantly more bitches neutered prepubertally had vulvas that changed to be smaller by 17 months of age, had smaller changes in vulval length and width between six and 17 months of age and had vulvas that appeared juvenile or recessed. This study considered vulval size and appearance prior to neutering and accounted for confounding variables in the analysis of vulval size.

The remaining 12 studies investigated the effects of timing of neutering related to age. Four of these made no comparisons of disease incidence between neuter age groups, only between each neuter age group and a group of entire dogs [15–17, 45]. Eight studies did make comparisons and three of those reported no impact of neutering age on urogenital disease [11, 26, 54].

Five studies reported effects of age at neutering on urogenital disease, specifically on recessed vulva [52]; cystitis and UI [12], and UI [47, 53, 58]. The only study to report more favourable results for bitches neutered earlier was a study examining UI onset post-neuter using data from veterinary records for 750 neutered bitches [58]. While the authors reported that the age at neuter and UI onset were related, with bitches neutered at older ages having earlier onset UI, they suggest that this "seems to confirm the lower risk in early spayed bitches compared to bitches spayed after the first oestrus, as reported by Stocklin-Gautschi *et al.* (2001)". However, the statistical evidence to support this statement is weak, with supporting data not presented in the main section of the results and no description of confounding factors included in statistical analysis.

The four papers that reported less favourable findings for bitches neutered at younger ages examined UI, cystitis and recessed vulva. Spain *et al.* [12] reported significantly higher incidence of cystitis in bitches neutered before 5.5 months of age and increasing incidence of UI with decreasing age at neuter, with the highest risk of UI for bitches neutered before three months of age. UI was only considered when bitches required medical treatment. The number of bitches in each neuter group was not stated. Dogs were of various breeds and crossbreeds, however pure/crossbreed status were included in the analysis as cofounding variables. It is not clear how reliable age at neuter determination was for the dogs that were in the shelter at the time of neutering. Byron *et al.* [47] also investigated UI and found a decreased hazard for UI in bitches neutered at older ages that was significant for dogs weighing over 25 kg. This unmatched case-control study did not consider, or exclude bitches based on, health prior to neutering. Dogs were recruited via invitations to contribute data that were sent to 500 veterinary hospitals in the USA. The lack of matching of case-control dogs and the lack of controlling for confounding variables such as breed in the analysis affect the risk of bias in the study. Pegram *et al.* [53] also identified a relationship between age at neuter and early-onset UI when data for 23,499 neutered bitches were examined. In that study neutering before six months of age was associated with an increased hazard of early-onset UI. Data for a large number of bitches from UK primary care practices were examined with confounding variables including breed and bodyweight included in the analysis.

Finally, Palerme *et al.* [52] reported that recessed vulva, determined by direct observation of lateral and dorsal vulvar skin fold coverage, was significantly more likely in bitches neutered at or less than 12 months or 24 months of age, compared to those neutered over these ages. However, no differences were identified when age at neuter was grouped as less or greater than four or eight months of age. The study included various breeds and bitches were recruited from a university veterinary medical centre. Simple statistical analyses were performed with no controlling for confounding variables. The method of obtaining thorough history is unclear, therefore it is not obvious whether the approximate age at the time of ovariohysterectomy used for analysis was based on owner recollection or from veterinary records.

The effects on bitch health of the timing of neutering in relation to puberty have been investigated in only a small number of studies. Other studies have suggested that neutering before or after puberty has been examined, however on closer examination, age at neuter and not pubertal status has been used to categorise neuter timing [11] or the reported ages at neuter for the prepubertal and post-pubertally neutered groups suggest a risk of bias [22].

In summary there are few studies that investigate the health implications of neutering bitches considering pubertal status at neutering. Most studies that do consider pubertal status investigate urogenital health, commonly UI, and are subject to a risk of bias. Studies generally propose that neutering bitches earlier appears to be protective for some (but not all) cancers but increases the risk of DOD and potentially UI; contrasting results which make applying the findings to benefit bitch health difficult.

The limitations of this scoping review are that the database searches were conducted in May 2023. Since that time, further studies may have become available. Additionally, the search strategy may not have identified all of the literature, however the databases used are reported to provide high levels of coverage of veterinary literature [44]. This review was completed by two individuals, which was deemed appropriate for the volume of literature generated for review. The search terms and strategy were reviewed by a group of five specialists, one with extensive experience of conducting scoping reviews. Additionally, the inclusion of a librarian helped to optimise the search strategy [28, 40, 68, 69] and papers were included that were found in a previous literature search by the primary author and did not appear in the search results. Therefore, it is more likely that the search would have been too broad than failing to identify relevant articles. The search term 'spey' was not included which could have resulted in articles being missed if they did not also include other terns relating to neutering that were in the search terms.

A scoping review rather than systematic methodology was chosen due to the broad question posed and the need to examine and describe the potentially large range of literature that had not previously been synthesised [38, 40], at least for three of the five health outcomes included. The methodology was appropriate to provide an overview of the available and relevant literature and to identify gaps in the research [39, 41, 70, 71] and meant that the papers related to age at neuter could be included and summarised, rather than just the small number of papers that considered puberty, as would have been considered in a systematic review. This provided a broader overview of the available evidence and identified problems relating to features of the study methodologies. This allowed reporting of the factors relating to study design and characteristics, and recommendations to be made for future studies. The review also identified that there is insufficient literature for systematic reviews to be undertaken for most of these diseases.

Future studies that aim to examine the effects of neutering bitches before or after puberty should:

- be undertaken for a variety of dogs (pet, assistance, shelter etc.) from many geographical locations,

- consider breed and body weight, which can impact health outcomes,

- clearly define inclusion criteria including whether diseases are diagnosed before or after the neutering intervention, which should also be defined,

- randomly allocate dogs to neutering groups where all bitches are neutered using the same surgical method,

- consider study population size in order to ensure that the minimum number of bitches needed for valid statistical analyses are recruited, considering the likely disease incidence,

- use power analysis to determine appropriate values of alpha for the study population,

- identify and define puberty, considering the onset of puberty rather than age.

For studies examining the effects of timing of neutering relating to age, previous breeding history should be considered. Additionally, consideration of other studies when selecting age groups could enable easier comparisons to be made between studies.

## Conclusion

This scoping review has identified a lack of evidence related to the impacts of neutering bitches before or after puberty on five aspects of bitch health (atopy, DOD, neoplasia, obesity and

urogenital disease). Only six studies were identified that categorised the timing of surgical neutering as prepubertal or post-pubertal, one investigating mammary neoplasia and the other five, urogenital disease, commonly UI. No studies were identified that examined the impacts on atopy, DOD or obesity. Twenty-six studies considered bitch age rather than pubertal status at the time of neutering. Most studies were retrospective and varied in their methodologies, quality and means of categorising bitches into neutering age groups. The review highlighted the lack of literature along with weaknesses in the existing studies caused by age rather than puberty being examined. Whether or not neutering before or after puberty has an impact on the health of female domesticated dogs is currently unknown.

## Supporting information

**S1 Checklist. PRISMA-ScR checklist.**
(DOCX)

**S1 File. Search terms.** The search terms used in a scoping review designed to identify and chart the current evidence on the effect of the timing of neutering in relation to puberty on health in female domesticated dogs.
(DOCX)

**S2 File. Inclusion and exclusion criteria.** The inclusion and exclusion criteria for a scoping review of the literature to identify and chart the current evidence on the effect of the timing of neutering in relation to puberty on the health of female domesticated dogs.
(DOCX)

**S3 File. Scoping review extraction form.**
(DOCX)

**S4 File. List of all studies identified in the searches and reasons for exclusion.**
(XLSX)

## Author Contributions

**Conceptualization:** Rachel Moxon, Gary C. W. England, Richard Payne, Sandra A. Corr, Sarah L. Freeman.

**Data curation:** Rachel Moxon, Sarah L. Freeman.

**Formal analysis:** Rachel Moxon, Sarah L. Freeman.

**Investigation:** Rachel Moxon, Sarah L. Freeman.

**Methodology:** Rachel Moxon, Gary C. W. England, Richard Payne, Sandra A. Corr, Sarah L. Freeman.

**Project administration:** Rachel Moxon.

**Supervision:** Rachel Moxon, Gary C. W. England, Richard Payne, Sandra A. Corr, Sarah L. Freeman.

**Visualization:** Rachel Moxon, Gary C. W. England, Richard Payne, Sandra A. Corr, Sarah L. Freeman.

**Writing – original draft:** Rachel Moxon.

**Writing – review & editing:** Rachel Moxon, Gary C. W. England, Richard Payne, Sandra A. Corr, Sarah L. Freeman.

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
