## [Decision Letter · Decision Letter 0]

8 Aug 2024

PONE-D-24-20315Effect of neutering timing in relation to puberty on health in the female dog - a scoping reviewPLOS ONE

Dear Dr. Moxon,

Thank you for submitting your manuscript to PLOS ONE. After careful consideration, we feel that it has merit but does not fully meet PLOS ONE’s publication criteria as it currently stands. Therefore, we invite you to submit a revised version of the manuscript that addresses the points raised during the review process.

We look forward to receiving your revised manuscript.

Kind regards,

Ioannis Savvas, DVM, Ph.D.

Academic Editor

PLOS ONE

Journal Requirements:

Reviewers' comments:

Reviewer's Responses to Questions

**Comments to the Author**

1. Is the manuscript technically sound, and do the data support the conclusions?

Reviewer #1: Yes

Reviewer #2: Yes

2. Has the statistical analysis been performed appropriately and rigorously? 

Reviewer #1: Yes

Reviewer #2: Yes

3. Have the authors made all data underlying the findings in their manuscript fully available?

Reviewer #1: Yes

Reviewer #2: Yes

4. Is the manuscript presented in an intelligible fashion and written in standard English?

Reviewer #1: Yes

Reviewer #2: Yes

5. Review Comments to the Author

Reviewer #1: In this scoping review, worthful details on studies dealing with the impact of neutering bitches before or after puberty on specific health issues are given, showing how difficult it is to compare these studies and that dogs were mostly not examined / puberty mostly not excluded. This is the highest benefit of this review. It is impressive that so far only 6 publications provided information about pubertal stage at neutering; in all others, neutering at a younger age may be more applicable. The provided scoping review appears technically sound and is presented in an intelligible fashion and standard English.

However, many of the here considered problems/diseases are multifactorial rendering reviews and especially meta-analyses very difficult as homogenous groups are difficult to find; neutering is probably only one factor leading to an aggravation of an already existing problem. I would really like to find a respective remark in the introduction, even though in the discussion it is well mentioned and elaborated.

Line 66: in the study of Schneider et al (1969), authors found a significant decrease in the relative risk of developing a mammary carcinoma, when female dogs were neutered before puberty. In most reviews I miss this clear differentiation between tumours/neoplasias in general and carcinomas. Even though the introduction shall only give an overview, it should be more differentiated.

The discussion is somewhat lengthy; many details are provided in the tables that are very helpful. Maybe authors can shorten the text.

I really like the proposals what should be considered in future studies, at the end of the discussion. The conclusions are feasible.

Line 466: …that had

Reviewer #2: Dear authors thank you for providing this scoping review for the investigation of the effect of neutering of female dogs before and after puberty on several health outcomes. This review was a nicely- written and interesting study that failed to identify any evidence concerning the effect of neutering on atopic dermatitis, neoplasia, obesity, DOD andurogenital disease. As such it warrants publication in PLOS ONE. My only comment was to change spey to spay in line 646.

6. PLOS authors have the option to publish the peer review history of their article (what does this mean?). If published, this will include your full peer review and any attached files.

Reviewer #1: No

Reviewer #2: No

---

## [Author Response · Author response to Decision Letter 0]

29 Aug 2024

Reviewer #1: In this scoping review, worthful details on studies dealing with the impact of neutering bitches before or after puberty on specific health issues are given, showing how difficult it is to compare these studies and that dogs were mostly not examined / puberty mostly not excluded. This is the highest benefit of this review. It is impressive that so far only 6 publications provided information about pubertal stage at neutering; in all others, neutering at a younger age may be more applicable. The provided scoping review appears technically sound and is presented in an intelligible fashion and standard English.

Response: Thank you very much for reviewing the manuscript and for your comments.

However, many of the here considered problems/diseases are multifactorial rendering reviews and especially meta-analyses very difficult as homogenous groups are difficult to find; neutering is probably only one factor leading to an aggravation of an already existing problem. I would really like to find a respective remark in the introduction, even though in the discussion it is well mentioned and elaborated.

Response: Thank you for your suggestion. The following sentence in the first paragraph of the introduction has been extended to read “However, comparing and consolidating evidence from different studies is challenging due to variations in study populations and methodological approaches, along with the multifactorial nature of the diseases studied.” We hope that this addition suitably acknowledges this point in the introduction.

Line 66: in the study of Schneider et al (1969), authors found a significant decrease in the relative risk of developing a mammary carcinoma, when female dogs were neutered before puberty. In most reviews I miss this clear differentiation between tumours/neoplasias in general and carcinomas. Even though the introduction shall only give an overview, it should be more differentiated. 

Response: Thank you for identifying this oversight, and we agree that this is commonly not differentiated in many reviews. This sentence has been altered so that the cancers studied are clearly presented. This now reads “Most reviews reference the same work by Schneider et al. [20] that identified a reduced risk of histologically malignant mammary neoplasms (adenocarcinomas and mixed mammary tumours) for bitches neutered before the first oestrus.”

The discussion is somewhat lengthy; many details are provided in the tables that are very helpful. Maybe authors can shorten the text. 

Response: Thank you for your suggestion. We have reviewed the discussion and we have removed a small amount of text and referred readers to Table 1 at one point. However, after reviewing the discussion, we think that the discussion relative to each category of disease, while for some categories is long, is useful to the reader to bring together a summary of the papers and related information in one place. For this reason, combined with the comments from the second reviewer, we decided to leave these sections of the discussion unchanged.

I really like the proposals what should be considered in future studies, at the end of the discussion. The conclusions are feasible.

Response: Thank you

Line 466: …that had

Response: This has been corrected, thank you

Reviewer #2: Dear authors thank you for providing this scoping review for the investigation of the effect of neutering of female dogs before and after puberty on several health outcomes. This review was a nicely- written and interesting study that failed to identify any evidence concerning the effect of neutering on atopic dermatitis, neoplasia, obesity, DOD and urogenital disease. As such it warrants publication in PLOS ONE. 

Response: Thank you very much for reviewing the manuscript and for your comments.

My only comment was to change spey to spay in line 646.

Response: Thank you for the suggestion, however the misspelling was intentional. The term ‘spay’ was included in the search terms, however the term ‘spey’, was not. Therefore, no changes have been made and this remains in the limitations section of the manuscript.

---

## [Decision Letter · Decision Letter 1]

17 Sep 2024

Effect of neutering timing in relation to puberty on health in the female dog - a scoping review

PONE-D-24-20315R1

Dear Dr. Moxon,

We’re pleased to inform you that your manuscript has been judged scientifically suitable for publication and will be formally accepted for publication once it meets all outstanding technical requirements.

Kind regards,

Ioannis Savvas, DVM, Ph.D.

Academic Editor

PLOS ONE

Additional Editor Comments (optional):

Reviewers' comments:

Reviewer's Responses to Questions

**Comments to the Author**

1. If the authors have adequately addressed your comments raised in a previous round of review and you feel that this manuscript is now acceptable for publication, you may indicate that here to bypass the “Comments to the Author” section, enter your conflict of interest statement in the “Confidential to Editor” section, and submit your "Accept" recommendation.

Reviewer #1: All comments have been addressed

2. Is the manuscript technically sound, and do the data support the conclusions?

Reviewer #1: Yes

3. Has the statistical analysis been performed appropriately and rigorously? 

Reviewer #1: Yes

4. Have the authors made all data underlying the findings in their manuscript fully available?

Reviewer #1: Yes

5. Is the manuscript presented in an intelligible fashion and written in standard English?

Reviewer #1: Yes

6. Review Comments to the Author

Reviewer #1: The review was well overworked, all concerns were addressed. The review provides important data; there is a severe lack on sound studies on the effect of prepubertal castration. As it is a scoping review, authors must evaluate the available literature in a strict manner, following rules. However, the practical aspect and the value of empirical data must not be forgotten. There will always be a little bias - for example it is impossible to fully exclude any tumour within the body before study begin - however, this was criticised in the Schneider et al (1969) study. So it is really important to rate the studies in a very differentiated manner. But as stated previously, i highly appreciate the listed parameters that should be considered in future studies, at the end of the discussion.

7. PLOS authors have the option to publish the peer review history of their article (what does this mean?). If published, this will include your full peer review and any attached files.

Reviewer #1: No

---

## [Editor Report · Acceptance letter]

4 Oct 2024

PONE-D-24-20315R1 

PLOS ONE

Dear Dr. Moxon, 

I'm pleased to inform you that your manuscript has been deemed suitable for publication in PLOS ONE. Congratulations! Your manuscript is now being handed over to our production team.

Kind regards, 

on behalf of

Prof. Ioannis Savvas 

Academic Editor

PLOS ONE